**AI-Assisted Spatio-Temporal Analysis of Forest Cover and Carbon Dynamics in**

**Northeast India:**

**A Remote Sensing and GIS Approach**

Arman Khan

Under the Guidance of Prof. Jainul Abudin

Department of Computer Science & Engineering

University of Science & Technology Meghalaya

15 May 2026

## Abstract

Northeast India is a globally recognised biodiversity hotspot facing severe ecological threats from rapid land-use and land-cover changes, including agricultural encroachment, urbanisation, and shifting cultivation. Traditional ground-based surveys are often prohibitively labour-intensive and inadequate for capturing macro-level landscape dynamics. This paper explores the integration of Geographic Information Systems, high-resolution multi-temporal satellite imagery, and machine learning classification methods to automate the spatio-temporal analysis of forest cover changes and carbon dynamics across the region. Drawing on recent regional data from Assam, Arunachal Pradesh, Manipur, and Mizoram, this study demonstrates how remote sensing and vegetation indices can accurately monitor environmental degradation, estimate aboveground biomass, and inform nature-based solutions for climate resilience.

*Keywords:* Artificial Intelligence, Carbon Dynamics, Change Detection, Geographic Information Systems (GIS), Land Use and Land Cover (LULC), Northeast India, Remote Sensing, Vegetation Indices.

Northeast India, nestled within the Eastern Himalaya and Indo-Burma biodiversity hotspots, is an epicentre of ecological wealth and indigenous knowledge (Behera, 2025). However, the region is highly vulnerable to climate change, land degradation, and resource-use conflicts. The primary drivers of deforestation and forest degradation in the region include agricultural and settlement encroachment, illegal resource extraction, infrastructure development, and shifting cultivation (TERI, 2024).

Monitoring these vast and often inaccessible terrains using traditional ground surveys is a labour-intensive and expensive process. To overcome these challenges, the integration of remote sensing and Geographic Information Systems has emerged as a crucial methodology. Furthermore, integrating artificial intelligence and machine learning algorithms into these spatial analyses provides a robust, scalable, and cost-effective approach to automating land-use and land-cover classification, monitoring biodiversity, and predicting carbon stock dynamics (Patil et al., 2024).

**Methodology**

Evaluating ecological changes requires multi-temporal satellite data and advanced analytical indices. Recent studies in the region have successfully utilised imagery from Landsat 7, Landsat 8, and Sentinel-2 to map landscape transformations (Borgohain & Bora, 2025; Lalhmachhuana et al., 2022). To automate and enhance the accuracy of land cover classification, researchers employ machine learning methods such as the Support Vector Machine, a supervised classification algorithm that processes satellite imagery pixel-by-pixel (Patil et al., 2024).

This pixel analysis is augmented by the calculation of specialised vegetation indices to determine plant health, soil conditions, and green canopy density. For example, the Normalised Difference Vegetation Index quantifies general vegetation greenness and health by measuring the difference between near-infrared and red bands (Borgohain & Bora, 2025). Additionally, indices such as the Soil Adjusted Vegetation Index and Modified Soil Adjusted Vegetation Index correct for soil brightness, making them highly effective for monitoring sparsely vegetated areas, forest degradation, and changing soil backgrounds (Nivia et al., 2026).

**Spatio-Temporal Analysis of Forest Cover Changes**

Artificial intelligence-assisted spatial analysis across several Northeast Indian states reveals distinct regional trends. In Arunachal Pradesh's Lower Dibang Valley, machine learning classification of Landsat data from 2009 to 2021 indicated significant ecological alterations, including an 8% decrease in forest area and a 6% decrease in rangeland and scrubland (Patil et al., 2024). These landscape changes independently impacted climatic variables such as temperature, precipitation, and specific humidity. In Assam, broader spatial analysis reveals that 10,076.43 square kilometres of forest experienced degradation between 2000 and 2022 (TERI, 2024). Localised studies confirm these trends; for instance, spatial analysis in Nameri National Park from 1988 to 2025 revealed a 7.86% decline in open forest cover and a 3.55% expansion in built-up areas, heavily driven by agricultural encroachment and the unregulated expansion of eco-tourism infrastructure along the park's periphery (Borah, 2025). Conversely, dense vegetation increased in specific zones of Assam's Lakhimpur district due to targeted tree plantations by forest committees and sustained canopy maintenance in regional tea estates (Borgohain & Bora, 2025).

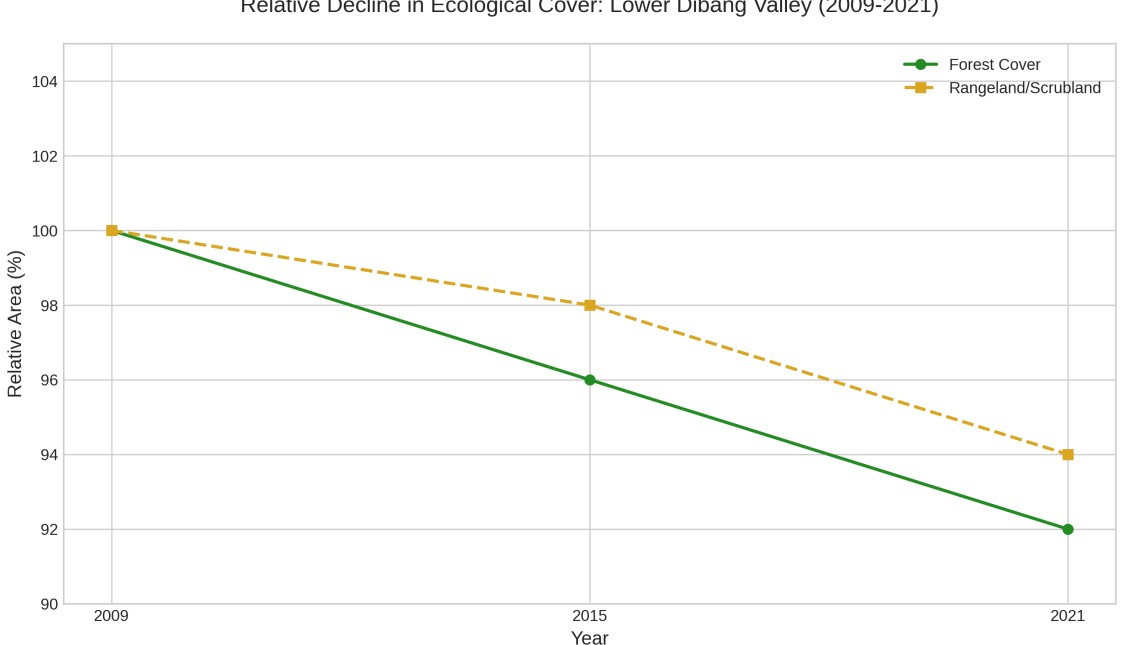

Similar transformations are evident in other states. Multi-spectral analysis utilising vegetation indices from 2018 to 2023 in Manipur highlighted rapid urban sprawl. The data revealed a substantial increase in bare soil, expanding from roughly 343 square kilometres in 2018 to 523 square kilometres in 2023, reflecting fast urbanisation and a severe loss of vegetated areas (Nivia et al., 2026). In contrast, a comparative analysis of satellite imagery from 2006 and 2012 in Aizawl, Mizoram, demonstrated that while built-up and agricultural areas increased, there was a substantial decrease in shifting cultivation areas (Lalhmachhuana et al., 2022). This decline in shifting cultivation, likely due to government-introduced permanent cultivation practices, correlated with a regional increase in forest cover.

**Carbon Stock Dynamics and Climate Resilience**

The integration of spatial intelligence extends beyond mapping to the critical assessment of regional carbon pools. Forests act as terrestrial carbon sinks, sequestering carbon in aboveground biomass, belowground roots, and soil. Satellite-derived vegetation indices are currently utilised to calculate tree volume and aboveground biomass. Studies synthesising biomass data across Northeast India highlight an average total biomass carbon estimate of 136.86 megagrams per hectare, with aboveground biomass contributing the vast majority (Dasgupta & Das, 2025). Standardised benchmarks commonly estimate that carbon constitutes approximately 47% to 50% of dry living biomass in these tropical ecosystems.

Crucially, the highest biomass and carbon sequestration rates are recorded in traditionally protected areas, such as the sacred groves of Manipur and Meghalaya, underscoring the effectiveness of indigenous conservation practices in maintaining old-growth trees and ecosystem stability (Dasgupta & Das, 2025). Addressing deforestation drivers requires scaling nature-based solutions, which are actions that protect, restore, and sustainably manage ecosystems using geospatial intelligence. For example, remote sensing is actively used to identify recharge zones for aquifer-based spring-shed management and to monitor wetland restoration efforts for flood resilience (Behera, 2025).

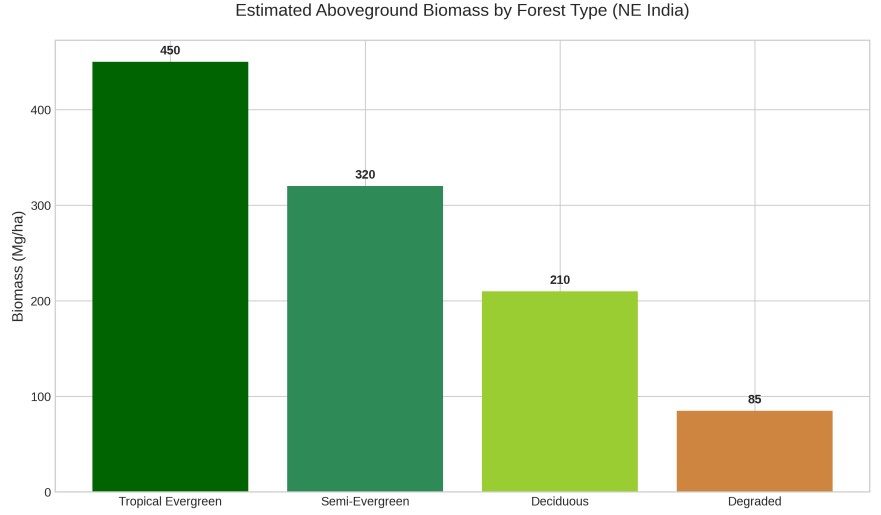

**Conclusion**

The integration of machine learning and multi-temporal satellite imagery provides an indispensable framework for monitoring forest degradation, urban sprawl, and carbon dynamics in Northeast India. The data confirms that while commercial plantations and eco-tourism offer economic benefits, they frequently drive habitat fragmentation if left unregulated. To achieve sustainable development goals, policymakers must utilise geospatial artificial intelligence tools to identify ecological hotspots, support nature-based solutions, and foster community-led conservation. Future research must focus on harmonising biomass estimation models and deploying advanced machine learning techniques to further refine real-time carbon assessments across these diverse terrains.

**AI Transparency Statement**

In accordance with the NortheastGenAI 2026 open experiment guidelines, this draft was collaboratively conceptualized, structured, and drafted using an AI assistant (MWire Labs, 2026). The AI was utilized to rapidly aggregate, synthesize, and format recent land-use, remote sensing, and biomass data from multiple imported academic sources into a coherent draft optimized for the thematic requirements of Track 3. All primary data points, analytical methods, and regional findings presented herein reflect the original sourced materials. As the human author, I determined the research direction, selected the specific track, verified the regional data points against the primary sources, and provided final editorial oversight before submission.

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

on Forest and Biodiversity Conservation Society.