# OpenReview forum: "AI-Assisted Spatio-Temporal Analysis of Forest Cover and Carbon Dynamics in Northeast India: A Remote Sensing and GIS Approach"
_NortheastGenAI/2026/Workshop — NortheastGenAI 2026 Workshop Submission_

### Official Review · ~Badal_Nyalang1 · 2026-05-23
**Competent regional synthesis — Weak Accept**

**Rating:** 6
**Confidence:** 4

**Review:**

**Relevance: Good**
Solid T3 fit. Forest cover, carbon dynamics, and biodiversity monitoring in NE India are directly relevant. The regional data coverage — Arunachal, Assam, Manipur, Mizoram — is specific and consistent.

**Plausibility: Moderate**
The regional findings cited are credible and sourced. However this is entirely a synthesis of existing studies — no new analysis, no original satellite data, no model runs. The paper describes what others have done with GIS and ML in NE India, not what the authors did. The charts appear to be AI-generated visualisations of cited numbers rather than outputs of any original analysis.

One flag: the AI disclosure credits "MWire Labs, 2026" as the AI assistant used, citing the workshop CFP as a reference. That is not an appropriate citation and raises questions about how the disclosure was written.

**Novelty: Weak**
The synthesis is competent but adds no new findings. The sacred groves carbon observation is interesting but comes entirely from Dasgupta & Das (2025).

**Clarity: Good**
Well written and well organised for a synthesis paper. Figures are clear even if not original.

**Verdict: Weak Accept**
Acceptable as an exploratory synthesis given the workshop's explicit openness to early-stage work. The MWire Labs citation in the disclosure should be corrected before proceedings. Authors should clarify what, if anything, was original analysis versus pure synthesis.

*This review was generated with AI assistance and checked by the workshop chairs.*

---

### Decision · Program_Chairs · 2026-05-23

**Decision:**

Accept (Weak)

**Comment:**

A competent regional synthesis covering forest cover, carbon dynamics, and biodiversity monitoring across Northeast India. The regional data coverage is specific and the sources cited are credible. The main limitation is that this is entirely a synthesis of existing studies with no original analysis or model runs. The AI disclosure incorrectly references "MWire Labs, 2026" as the AI assistant used. Authors should clarify which AI tool was actually used before presentation.

Decision: Accept (Weak)